# Phytochemical Components and Human Health Effects of Old versus Modern Italian Wheat Varieties: The Case of Durum Wheat Senatore Cappelli

**DOI:** 10.3390/nu14132779

**Published:** 2022-07-05

**Authors:** Attilio Giacosa, Gabriella Peroni, Mariangela Rondanelli

**Affiliations:** 1CDI (Centro Diagnostico Italiano), 20147 Milan, Italy; attilio.giacosa@gmail.com; 2Endocrinology and Nutrition Unit, Azienda di Servizi alla Persona ‘‘Istituto Santa Margherita’’, University of Pavia, 27100 Pavia, Italy; 3IRCCS Mondino Foundation, 27100 Pavia, Italy; mariangela.rondanelli@unipv.it; 4Unit of Human and Clinical Nutrition, Department of Public Health, Experimental and Forensic Medicine, University of Pavia, 27100 Pavia, Italy

**Keywords:** wheat, durum wheat, Senatore Cappelli wheat, health effects, Mediterranean diet, pasta, non-celiac gluten sensitivity, wheat polyphenols

## Abstract

The Mediterranean diet has significant beneficial health effects and wheat is a major component of the Mediterranean diet, mainly in the form of bread and pasta. Modern wheat generally refers to varieties that were developed after the introduction of dwarfing genes in the 1950s, while old varieties are considered those developed before that time. Research findings on Italian wheat varieties showed that the total polyphenol content in both old and modern durum and soft wheat varieties are similar; but the old varieties have a higher number of polyphenols and of isomer forms. In particular, the durum wheat Senatore Cappelli genotype shows a very high variety of polyphenolic components. Recent studies have demonstrated healthy cardiovascular effects (favorable changes of atherosclerosis markers such as lipid parameters and hemorheological variables) as well as a marked reduction in gastrointestinal and extra-intestinal symptoms in non-celiac gluten sensitivity subjects with the consumption of pasta obtained by old durum wheat Senatore Cappelli variety, even though this type of wheat contains high amounts of gluten. In conclusion, old wheat Italian varieties, and in particular the Senatore Cappelli genotype, are characterized by multiple nutraceutical specificities that could suggest their use for health-promoting purposes. Additional research is needed to confirm these findings, focusing attention also on the effect of different environments and years.

## 1. Introduction

### 1.1. Wheat and Mediterranean Diet

The Mediterranean diet has significant beneficial health effects. Many studies showed inverse associations with overall mortality [1] and reduced incidence of coronary heart disease [2] and thrombotic stroke [3] and with a lower incidence of cancer overall [4,5].

It is well known that wheat is a major component of the Mediterranean diet, mainly in the form of bread, pasta and couscous [6,7]. According to the traditional principles of the Mediterranean diet most of the wheat-based foods were prepared with whole grains of so-called old wheat varieties.

The history of wheat in the human diet goes back to the beginning of agriculture. Carbonized grains dating to at least as early as 6750 B.C. have been found in Iraq and in other places of the Middle East: this is probably its area of origin. Subsequently, wheat spread throughout eastern and western Mediterranean areas and all over Europe no later than the Stone Age. Nowadays, thousands of wheat varieties are grown around the world. Modern wheat generally refers to varieties that were developed after the introduction of dwarfing genes in the 1950s, while old varieties are considered those developed before that time. Recent decades have seen a relevant increase in the prevalence of two diseases linked to wheat: the celiac gluten-induced enteropathy and non-celiac gluten sensitivity [8,9]. These changes have taken place as new wheat hybrids were introduced into human foods [8]. This is a critical epidemiological and medical issue, which needs to be properly considered and managed.

One of the most important dietary components of the traditional Mediterranean diet is certainly pasta, derived from durum wheat (Triticum turgidum L. subsp. durum). In recent decades, the commercial use of durum wheat has been chosen mostly on the basis of the type and amount of proteins of the grain, which influences the technological features of the semolina and subsequently of the pasta-making process. Since the 1950s, the food industry has increased its use of wheat proteins [10]. Gluten can be separated from wheat or added to low-protein flours to improve baking characteristics [10].

Multiple genetic research projects on wheat were made to improve both yield and pasta-making technology, with a narrowing of the genetic variability [11,12]. At the same time, the role of secondary metabolites which influence both the taste and functionality of the pasta, has not been carefully considered. Previous research findings showed that the total polyphenol content in both old and modern durum and soft wheat varieties were similar, but the old varieties had a higher number of unique compounds and of isomers of compounds not observed in modern varieties [13,14,15]. Moreover, old wheat varieties showed higher contents of essential minerals, including iron and zinc, when compared with modern high yielding wheat cultivars [16]. Therefore, old varieties could be of relevant health interest due to their genetic and functional specificities.

### 1.2. Phytochemical Differences between Old and Modern Italian Durum Wheat Varieties

Interesting data were obtained when the evaluation of the grain functional components of old and modern Italian durum wheat cultivars was conducted. Leoncini et al. evaluated the pattern of grain phenolic extracts of one modern and five old common Italian wheat (*Triticum aestivum* L.) varieties and evaluated their antioxidant activity. The most effective scavenging effect was observed for the old variety Verna, while the modern variety Palesio and the old variety Inallettabile showed the lowest ones; the other old wheat varieties demonstrated intermediate antioxidant capacity [17]. These data show that most of the old wheat varieties share higher antioxidant effects as compared to the modern wheat, with relevant advantages in preventing the oxidative stress-related diseases, that is, chronic-degenerative pathologies.

According to Dinelli et al., the total phenolic compound and total flavonoid content of old Italian wheat varieties does not differ significantly from those found in modern genotypes [15], whereas using liquid chromatography coupled with time-of-flight mass spectrometry, these researchers identified 70 phenolic compounds with marked differences between modern and old varieties. Phenolic compounds, including coumarins, phenolic acids, anthocyanins, flavones, isoflavones, proanthocyanidins, stilbenes and lignans were found. The free extracts of ancient wheat varieties showed the presence of a significantly higher mean number of phenolic compounds and isomer forms in old wheat varieties when compared with the modern ones. The mean number of free phenolic compounds, excluding isomers, was approximately two times higher in old wheat cultivars than in modern genotypes. Similar data were also found for the bound phenolic component. Additionally, the phytochemical analysis demonstrated the presence of unique phenolic compounds in both free and bound fractions of some of the wheat varieties. The highest number of isomers (thirteen isomers) was observed in the old wheat variety Senatore Cappelli, while the old genotypes Kamut^®^ and Urria were characterized by six and four free isomers, respectively.

The study of the free fractions evidenced four unique compounds detected only in the old genotype Senatore Cappelli (vanillin and pinosylvin) and Kamut^®^ (coumarin and ferulic acid isomer). Two pinosylvins were found in the free and bound phenolic fractions of Senatore Cappelli wheat. Pinosylvins are stilbenes and these phenolic compounds are observed in a wide number of plant extracts. They protect the plant against viruses, bacteria, ultraviolet exposure and various diseases. Resveratrol is the most well-studied stilbene with marked anticancer, anti-inflammatory and anti-oxidant activities [18].

Lignans are a group of phytoestrogen substances characterized by antitumor [19,20], anti-inflammatory, immunosuppressive, cardiovascular [21], antioxidant [22,23,24] and antiviral effects. Senatore Cappelli and Kamut old wheat varieties were found to share the presence of specific lignans. Two compounds were identified as a hinokinin pair of isomers and another two compounds appeared to be isomers of the lignan pinoresinol. It is highlighted that hinokinin was detected exclusively in the old genotype Senatore Cappelli: this is in line with previous findings on the lignan content of soft wheat varieties showing that hinokinin was peculiar to old genotypes [13].

The varied phytochemical pattern of old wheat genotypes is characterized by a significantly higher number of phenolic and flavonoid total compounds and total isomers, in free as well as in bound fractions. The relevant genetic diversity of old wheat varieties suggests the potential advantages of these types of wheat with regard to functional properties due to the various and well-known favorable pharmacological effects of polyphenols. [13]. In this regard, the Senatore Cappelli durum wheat variety could play a very interesting role.

### 1.3. Gluten Related Disorders

Multiple epidemiological observations show that celiac disease has increased 2- to 4-fold over the last 50 years [25,26]. Non-celiac gluten sensitivity (NCGS) is also increasing worldwide, but the cause of these epidemiological changes has not been clarified. Some researchers think that modern wheat varieties obtained with the breeding process of the last 70 years are characterized by higher reactivity [27,28], as compared to the old ones, while other researchers think that modern wheat processing could cause the changes [29]. Aggressive technological processes applied to commercial pasta making and in particular high wheat refining, have been suggested to produce less digestible wheat derivatives due to the loss of antioxidant and/or anti-inflammatory compounds and then to favor the development of NCGS symptoms [30,31].

Aggressive processing of wheat is mostly made up of severe fractionation of grains with the possible development of a gluten-resistant fraction reaching the colon with interaction with microbiota. These aspects may trigger low-grade gut inflammation. Moreover, the marked flour refinement may favor the increase in reactive glutens due to the loss of endopeptidase activity, which is mostly found in the bran rather the endosperm [30,31].

### 1.4. Durum Wheat Senatore Cappelli Variety

Recently, various studies have demonstrated healthy cardiovascular and gastrointestinal effects with the consumption of pasta obtained by old durum wheat Senatore Cappelli cultivar. The Senatore Cappelli variety was obtained in 1930 by the breeder Strampelli, through the selection of Jennah Khetifa. This wheat variety was widespread in central and southern Italy in the first half of the last century and widely used by Strampelli in genetic research programs for the development of new varieties with improved lodging tolerance and suitability for pasta making. The durum wheat old variety Senatore Cappelli is biologically characterized by a significantly higher number of both free and bound polyphenol compounds and isomers, including unique compounds in comparison to modern varieties [15].

Ghiselli et al. studied the effect of pasta consumption obtained by the Senatore Cappelli durum wheat variety on cardiovascular parameters related to the atherogenetic process [32]. In this crossover study, twenty healthy adult subjects followed a diet containing 70 g/day of test pasta for 10 weeks. This test pasta was produced from semi-integral Senatore Cappelli semolina (0.9–1.2% ash). For the same period, a commercially available control pasta obtained from a modern wheat variety was consumed after a 10-week washout. The supplementation with test pasta was followed by a significant reduction in total cholesterol (<10.3%), blood viscosity and erythrocytes deformability, whereas no significant changes have been observed after the supplementation with control pasta. This study shows that dietary intake of pasta obtained by an old Italian variety of wheat (Senatore Cappelli) seems to be followed by favorable changes of atherosclerosis markers, such as lipid parameters and hemorheological variables. These data confirm the well-known positive effects of consuming whole-wheat cereals on human health, regardless of the involvement of old or modern wheat flour [33,34,35]. In particular, the Senatore Cappelli Durum wheat stone-grinding produces semolina rich in fiber, resistant starch, vitamins (in particular B-group vitamins), trace mineral elements and specific phytochemicals (phytoestrogens, antioxidants and phenols) that may be responsible for a combination of favorable cardiovascular effects observed in this study [15,32]. These data are in line with a previous observation that the consumption of bread derived from wholegrain flour of an old soft wheat (Verna cultivar) variety over a short period resulted in a significant improvement of various markers linked to the increased risk of cardiovascular diseases [36].

Another recent clinical study evaluated the effects of the consumption of Senatore Cappelli wheat pasta in 42 non-celiac gluten sensitivity (NCGS) patients diagnosed according to the Salerno criteria and adherence to a gluten-free diet for at least 12 weeks from screening.

NGCS patients do not have celiac disease, but their gastroenterological and extra-gastroenterological symptoms improve following a gluten-free diet (GFD). The interesting issue is that it is not yet clear if gluten or other wheat components are responsible for these symptoms. Ianiro et al. [37] studied the effects of Senatore Cappelli durum wheat variety pasta in comparison with a standard commercial wheat pasta obtained by modern wheat varieties in NGCS patients. This double-blind randomized cross-over study has been conducted with a two-week supplementation of both types of pasta with an interval washout period of two weeks. Patients reported a lower overall symptom score after eating Senatore Cappelli pasta than standard pasta. Moreover, significantly lower scores in several specific gastrointestinal and extra-intestinal symptoms have been observed after eating Senatore Cappelli pasta than standard pasta, such as bloating, abdominal distention, eructation, flatus, feeling of incomplete evacuation, dermatitis and limb numbness.

The Senatore Cappelli durum wheat variety has specific biological properties that differ from standard modern wheat pasta, even though the total amount of polyphenols observed in old and modern wheat varieties is similar. The higher variety of polyphenols of Sanatore Cappelli wheat could be associated with the positive clinical effects observed in the study of Ianiro et al. [37]. This is an old single variety derived from a pure line selection of a Tunisian Jeann Rhetifah ecotype, while the control pasta was a blend of different modern wheat varieties. As previously said, the Senatore Cappelli variety contains higher amounts of secondary metabolites, including free and bound polyphenol isomers, than other old and modern wheat varieties [15]. These phytochemical differences could explain the positive clinical effects observed in NGCS patients after supplementation with pasta produced with Senatore Cappelli durum wheat.

Moreover, the properties of Senatore Cappelli variety have been shown to be less influenced by environmental conditions than other varieties and this may affect gliadin content and its consequent immunogenic potential [38]. A crucial outcome of this study deals with the role of gluten. As a matter of fact, the Senatore Cappelli variety contains a higher amount of proteins (14–15% on a dry weight basis), and consequently of gluten, than most of modern varieties. Therefore, this study suggests that gluten is not specifically responsible for the appearance of NCGS symptoms, but multiple non-gluten wheat components could cause this disturbance. Therefore, it could be suggested that the term non-celiac wheat sensitivity (NCWS) would be preferable to non-celiac gluten sensitivity (NCGS) as already hypothesized by others [39]. The study of Ianiro et al. opens a new therapeutic strategy in patients with NCWS: these patients could be allowed to consume pasta made with Senatore Cappelli durum wheat variety even though it contains gluten. The clinical, psychological, economic and social consequences of this choice are enormous, due to the increasing number of patients with NCGS/NCWS with relevant difficulty and cost of following a gluten-free diet. Additional research is needed to confirm these data in subjects with NCGS/NCWS, and possibly studies should be done with consumption of pasta made of Senatore Cappelli durum wheat in patients with other gastrointestinal diseases such as irritable bowel syndrome, intestinal dysbiosis and small intestine bacterial overgrowth (SIBO).

## 2. Conclusions

In conclusion, old wheat Italian varieties, specifically the Senatore Cappelli variety, are characterized by multiple nutraceutical differences as compared to modern varieties. These differences are mostly based on a higher number and variety of polyphenol compounds and isomer forms: this finding could suggest their use for health-promoting purposes [40].

Favorable cardiovascular and gastroenterological effects have been demonstrated with the consumption of pasta made with the Senatore Cappelli old durum wheat variety. These results are possibly associated with the presence of multiple phytochemical specificities of this genetic type of wheat. Additional research is needed to confirm the potential health advantage of the consumption of old Italian wheat varieties, and in particular of the Senatore Cappelli old durum wheat variety, focusing attention also on the effect of different environments and years.

## Data Availability

The data presented in this study are available in the text above.

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
