# Peer review of "Phytochemical Components and Human Health Effects of Old versus Modern Italian Wheat Varieties: The Case of Durum Wheat Senatore Cappelli"

_nutrients, 2022, doi:10.3390/nu14132779_

Round 1
Reviewer 1 Report
The manuscript, "Phytochemical components and human health effects of old versus modern Italian wheat varieties: the case of durum wheat Senatore Cappelli," looked to characterize new versus old varieties of a type of wheat in senatore cappelli, how these varieties influence health and disease, and distinguished phytochemical compositions in between as a reflection of their benefits.
The first and second paragraphs in section 1.2. conflict, specifically relating to citation 15. Here, data suggest phytochemical composition between new and old wheat varieties do not differ significantly. The entire manuscript, based off the title, depends on such data as the argument seems to be that old varieties have better phytochemical composition and subsequent benefits than new varieties. The benefits may depend on other things. Nevertheless, more attention should be reflected here.
A table showing phytochemical compositions of new and old senatore cappelli would greatly benefit this manuscript. Also, please consider a figure that represents the many efficacious actions of individual phytochemicals found in senatore cappelli.
Author Response
June 27, 2022
To Editor-in-Chief,
We revised the manuscript with modifications and changes based on the reviewers’ comments.
We send you the revised manuscript together with our point-by-point response.
The changes in the text were written in red.
Thanking you in advance for your kind collaboration and suggestions.
Best regards,
The authors
Comments and Suggestions for Authors
The manuscript, "Phytochemical components and human health effects of old versus modern Italian wheat varieties: the case of durum wheat Senatore Cappelli," looked to characterize new versus old varieties of a type of wheat in senatore cappelli, how these varieties influence health and disease, and distinguished phytochemical compositions in between as a reflection of their benefits.
The first and second paragraphs in section 1.2. conflict, specifically relating to citation 15. Here, data suggest phytochemical composition between new and old wheat varieties do not differ significantly. The entire manuscript, based off the title, depends on such data as the argument seems to be that old varieties have better phytochemical composition and subsequent benefits than new varieties. The benefits may depend on other things. Nevertheless, more attention should be reflected here.
ANSWER: As a matter of fact, the first and second paragraphs in section 1.2 do not conflict because Dinelli (ref 15) reports that the total amount of polyphenols is similar in old and modern wheat varieties. What differs remarkably is the number of different components which appears much higher in the old ones. These results are perfectly in line with the higher scavenging effects observed in old varieties by Leoncini (ref 17).
In order to clarify properly the test, we include the term “whereas” in line 90.
A table showing phytochemical compositions of new and old senatore cappelli would greatly benefit this manuscript. Also, please consider a figure that represents the many efficacious actions of individual phytochemicals found in senatore cappelli.
ANSWER: We appreciate the suggestion of reviewer 1 to include a table and/or figure that focuses the attention on the differences of the phytochemical components between the old and modern wheat varieties, but this appears a repetition of what already reported in the text of Dinelli, table 5 and 6.
Reviewer 2 Report
This review described the health-promoting effects of “old” durum wheat Senatore Cappelli. It provides valuable information for the food industry and human health. Some technical comments are as follows:
Line 87-95: authors are suggested to show the structure of these phenolic compounds that are unique in the old variety.
Line 154-155: This claim is not appropriate here. Modern wheat variety can also be whole grain But in the literature authors cited, only the old variety of whole grain showed health benefits.
Line 181-182: Should clarify this. As authors mentioned, the old variety had more types of phenolics. The amounts seemed confusing.
Line 206-207: The authors summarize evidence supporting the additional benefits of wheat Senatore Cappelli, but not all old Italian wheat varieties. Therefore, this sentence should be changed.
Line 208-209: In the main text, authors also mention the difference in fiber and vitamin contents. Therefore, the contribution of fiber/vitamins should be acknowledged.
Line26-27/213-215: Authors are suggested to more clearly clarify that the health statement about Senatore Cappelli is tentative based on some current data. More experiments( such as Senatore Cappelli harvested in different enviroments/years or more human intervention studies) are needed.
Author Response
This review described the health-promoting effects of “old” durum wheat Senatore Cappelli. It provides valuable information for the food industry and human health. Some technical comments are as follows:
Line 87-95: authors are suggested to show the structure of these phenolic compounds that are unique in the old variety.
ANSWER: in this review we summarized the most relevant information reported in Dinelli’s paper and the structure of the phenolics compounds that mostly characterize the senator Cappelli variety are reported in the following paragraphs (line 97 – 112)
Line 154-155: This claim is not appropriate here. Modern wheat variety can also be whole grain But in the literature authors cited, only the old variety of whole grain showed health benefits.
ANSWER: we agree with the kind suggestion and we clarified the sentence with a new phrase: “These data confirm the well-known positive effects of consuming whole-wheat cereals on human health, regardless of the involvement of old or modern wheat flour”.
Line 181-182: Should clarify this. As authors mentioned, the old variety had more types of phenolics. The amounts seemed confusing.
ANSWER: this comment is similar to question 1 of reviewer 1: we reported the data of Dinelli’s paper showing that the total amount of polyphenols is similar in old and modern wheat varieties. What differs remarkably is the number of different components which appears much higher in the old ones.
The sentence has been changed: “The Senatore Cappelli durum wheat variety has specific biological properties that differ from standard modern wheat pasta, even though the total amount of polyphenols observed in old and modern wheat varieties is similar. The higher variety of polyphenols of Sanatore Cappelli wheat could be associated with the positive clinical effects observed in the study of Ianiro et al.”.
Line 206-207: The authors summarize evidence supporting the additional benefits of wheat Senatore Cappelli, but not all old Italian wheat varieties. Therefore, this sentence should be changed.
ANSWER: this paper is focused on the clinical benefits of Senatore Cappelli wheat consumption due to the existence of various controlled randomized clinical trials with this type of wheat while similar experiments with other old wheat varieties are not available.
Line 208-209: In the main text, authors also mention the difference in fiber and vitamin contents. Therefore, the contribution of fiber/vitamins should be acknowledged.
ANSWER: the potential benefits of the fiber and vitamins content of Senatore Cappelli wheat have been described in LINE 156-157 (REF 15 AND 32).
Line26-27/213-215: Authors are suggested to more clearly clarify that the health statement about Senatore Cappelli is tentative based on some current data. More experiments( such as Senatore Cappelli harvested in different enviroments/years or more human intervention studies) are needed.
ANSWER: according to the indications, we modified the text adding to abstract and conclusions this sentence: “focusing also the attention on the effect of different environment and year”
Reviewer 3 Report
Authors discussed phytochemical differences between old and modern italian durum wheat varieties, and their human health effects. However, I have some questions/comments on some parts.
1. Why the authors only discuss one modern wheat variety, is there any other important modern wheat variety in Italy?
2.Line 127-130, why aggressive technological processes produce less digestible wheat derivatives due to the loss of antioxidant and/or anti-inflammatory compounds ? please provide detail information or explaination.
3.Line 142-162, it would be better to provide the data of components ( fiber, resistant starch, vitamins, trace mineral elements and specific phytochemicals) in control and test pasta.
4.Line 163-176, the detail information of double-blind randomized cross-over study need to be provided.
5. Tables or figures are suggested to summarize previous studies on human health effects of old versus modern Italian wheat varieties and the possible machanisms.
Author Response
Authors discussed phytochemical differences between old and modern italian durum wheat varieties, and their human health effects. However, I have some questions/comments on some parts.
- Why the authors only discuss one modern wheat variety, is there any other important modern wheat variety in Italy?
ANSWER this paper focuses the attention on senator Cappelli old wheat variety, which has been the most used old Italian variety until the 50’s and represents the typical wheat varieties of the original Mediterranean diet in south of Italy. in addition to this, pasta made of senator Cappelli wheat variety has been studied in many randomized clinical trials, while similar experiments with other types of old Italian wheat are not available.
2.Line 127-130, why aggressive technological processes produce less digestible wheat derivatives due to the loss of antioxidant and/or anti-inflammatory compounds ? please provide detail information or explaination.
ANSWER: As reported by Kucek, L.K. et al. and Fardet et al the impact of the aggressive technological processing on wheat sensitivity are based on the following concepts:
Drastic processing in wheat technology are mainly grain fractionation and refining followed by recombination and salt, sugars and fats addition, being able to render ultra-processed cereal-based foods more prone to trigger chronic low-grade inflammation. Concerning bread, intensive kneading and the choice of wheat varieties with high baking quality may have rendered gluten less digestible, moving digestion from pancreatic to intestinal proteases. The hypothesis of a gluten resistant fraction reaching colon and interacting with microflora is also considered in relation with increased inflammation. Besides, wheat flour refining removes fiber co-passenger which have potential anti-inflammatory property able to protect digestive epithelium.
Moreover, the amount of reactive glutens may change with the level of flour refinement. Most endopeptidase activity was found in the bran rather than the endosperm (Hartmann and others 2006; Schwalb and others 2012). This distribution is not surprising, as cysteine proteases are synthesized in the aleurone layer of barley (Hammerton and Ho 1986). Because the bran is removed in the process of making white flour, subsequent products would have fewer enzymes available for prolamin degradation.
A new sentence has been added in the text.
Aggressive processing of wheat is mostly made up of severe fractionation of grains with possible development of a gluten resistant fraction reaching the colon with interaction with microbiota. These aspects may trigger low-grade gut inflammation. Moreover, the marked flour refinement may favor the increase of reactive glutens due to the loss of endopeptidase activity which is mostly found in the bran rather the endosperm.
3.Line 142-162, it would be better to provide the data of components ( fiber, resistant starch, vitamins, trace mineral elements and specific phytochemicals) in control and test pasta.
ANSWER: unfortunately, the data of components suggested by reviewer are not reported in the paper of Ghiselli et al. The only data provided are reported in table 1 of that paper and regard:
Mean and standard error (±SE) of 1000 Kernel Weight (1000 K.W.), Total Protein (T. Prot.), Total Carbon (T. C) and Gluten content.
4.Line 163-176, the detail information of double-blind randomized cross-over study need to be provided.
ANSWER: According to the reviewer’s suggestion, we added more detailed information about the study
- Tables or figures are suggested to summarize previous studies on human health effects of old versus modern Italian wheat varieties and the possible machanisms.
ANSWER: both the tables suggested have already been published by Dinu M et al (Ancient wheat species and human health: Biochemical and clinical implications, Journal of Nutritional Biochemistry 52 (2018) 1–9):
Table 1 of the paper: Contents of fiber and phytochemicals in old and modern wheat cultivars
Table 3 of the paper: Human intervention trials to evaluate the impact of wheat (ancient and modern cultivars) to improve risk factors related to nutrient-related chronic diseases
We added this reference in the conclusions.
Round 2
Reviewer 3 Report
The authors have answered all my questions.